# Enhancing the Relative Sensitivity of V^5+^, V^4+^ and V^3+^ Based Luminescent Thermometer by the Optimization of the Stoichiometry of Y_3_Al_5−x_Ga_x_O_12_ Nanocrystals

**DOI:** 10.3390/nano9101375

**Published:** 2019-09-25

**Authors:** Karolina Kniec, Karolina Ledwa, Lukasz Marciniak

**Affiliations:** Institute of Low Temperature and Structure Research, Polish Academy of Sciences, Okólna 2, 50-422 Wroclaw, Poland; k.ledwa@intibs.pl

**Keywords:** vanadium, gallium, garnets, inorganic nanocrystals, luminescence, luminescent nanothermometry

## Abstract

In this work the influence of the Ga^3+^ concentration on the luminescent properties and the abilities of the Y_3_Al_5−x_Ga_x_O_12_: V nanocrystals to noncontact temperature sensing were investigated. It was shown that the increase of the Ga^3+^ amount enables enhancement of V^4+^ emission intensity in respect to the V^3+^ and V^5+^ and thus modify the color of emission. The introduction of Ga^3+^ ions provides the appearance of the crystallographic sites, suitable for V^4+^ occupation. Consequently, the increase of V^4+^ amount facilitates V^5+^ → V^4+^ interionic energy transfer throughout the shortening of the distance between interacting ions. The opposite thermal dependence of V^4+^ and V^5+^ emission intensities enables to create the bandshape luminescent thermometr of the highest relative sensitivity of V-based luminescent thermometers reported up to date (S_max_, 2.64%/°C, for Y_3_Al_2_Ga_3_O_12_ at 0 °C). An approach of tuning the performance of Y_3_Al_5−x_Ga_x_O_12_: V nanocrystals to luminescent temperature sensing, including the spectral response, maximal relative sensitivity and usable temperature range, by the Ga^3+^ doping was presented and discussed.

## 1. Introduction

Inorganic nanocrystals, due to their high mechanical, thermal and chemical stability, have garnered an immense interest from the point of view of their potential implementation in biomedical application, i.e., optical and magnetic resonance imaging, drug delivery, light-induced hyperthermia generation etc. [1,2,3,4]. Their optical properties may be in a facile way modified by the introduction of the appropriate optically active ions like lanthanide (Ln^3+^) and/or transition metals (TM) ions [5,6,7,8,9,10,11,12,13] to the host material. Besides unique chemical and physical features, they reveal size- and shape-dependent spectroscopic properties, which are not observed for organic-based nanomaterials [1]. Due to the fact that the optical properties of such nanoparticles are strongly affected by the temperature, their luminescence may be employed to non-contact temperature sensing (luminescent thermometry, LT). In LT, temperature readout relies on the analysis of thermally-affected spectroscopic parameters like emission intensity, luminescence lifetime, peak position, band shape and polarization anisotropy [14,15,16]. One of the most important advantages of LT in respect to other temperature measurement techniques is the fact that it provides a real-time temperature readout with unprecedented spatial and thermal resolution [15,17,18]. Additionally, temperature readout is provided in an electrically passive mode what enables to achieve the information about, i.e., the condition of living organisms where even small temperature fluctuations are usually accompanied by serious health diseases and improper cellular biochemical processes [16,19,20,21,22]. The use of the nanosized LTs enables the improvement of the spatial resolution of temperature readout. However, in order to obtain high thermal resolution of temperature measurement, different approaches, which enable to increase the relative sensitivity of LT to temperature changes, were proposed up to date. As was recently demonstrated, the utilization of transition metal ions luminescence with lanthanide co-dopant as a luminescent reference enables the enhancement of temperature sensing sensitivity, luminescence brightness and the broadening of usable temperature range in which LT operates [23,24,25]. For this purpose, optical properties of different TM were investigated, such as V^3+^/V^4+^/V^5+^ [23,26], Co^2+^ [27], Ti^3+^/Ti^4+^ [28], Cr^3+^ [24,25], Mn^3+^/Mn^4+^ [29] and Ni^2+^ [30]. Another advantage of using TM is the susceptibility of their optical properties to the modification of the crystal field strength via host stoichiometry due to the fact that *d* electrons, located on the valence shell, are exposed to the local environment and crystal field changes. This phenomenon was investigated in detailed in case of temperature sensing performance of Cr^3+^ ions where the structure of host materials were varying from Gd_3_Al_5_O_12_ (GAG) to Gd_3_Ga_5_O_12_ (GGG), and from Y_3_Al_5_O_12_ (YAG) to Y_3_Ga_5_O_12_ (YGG) via changing the Al^3+^ to Ga^3+^ ratio [24,25]. As was recently shown for Cr^3+^ ions, such modification enables not only enhancement of the sensitivity of LT but also tuning of the spectral position of emission band [25]. These kinds of studies have not yet been conducted for V-based luminescent thermometers. 

Therefore, in this work, we present for the first time a strategy that enables the improvement of temperature-sensing properties of V-based luminescent nanothermometers via modification of the host material composition. This approach bases on the gradual substitution of Al^3+^ ions by Ga^3+^ ions into YAG nanocrystals. The introduction of gallium ions, which possess larger ionic radii in respect to Al^3+^ ones leads to the lowering of crystal field (CF) strength. This arises from the elongation of the metal-oxygen (M-O) distance along with the enhancement of the contribution of Ga^3+^ ions. The modification of the crystal field strength should strongly influence the temperature-dependent luminescent properties of V ions of different oxidation state (V^5+^, V^4+^, V^3+^). Moreover, the introduction of the gallium ions facilitates the stabilization of V^4+^ oxidation state that possesses favorable performance for luminescent thermometry. However, these expectations have not yet been experimentally verified. Therefore, the aim of this work is to study the influence of the Ga^3+^ ions concentration of the temperature dependent luminescent properties of vanadium ions in Y_3_Al_5−x_Ga_x_O_12_:V nanocrystals, with the special emphasize put on their application in luminescent thermometry.

## 2. Materials and Methods

### 2.1. Synthesis of V-doped Y_3_Al_5−x_Ga_x_O_12_

The Y_3_Al_5−x_Ga_x_O_12_ nanocrystals doped with 0.1% concentration of V ions were synthesized via a modified Pechini method, where the Ga^3+^ amount was set to x = 1, 2, 3, 4 and 5. The amount of V ions was set to 0.1% due to the fact that this V concentration provides the most significant temperature sensing properties of YAG:V, Ln^3+^ luminescent nanothermometers [23]. The first step was the creation of yttrium nitrate from yttrium oxide (Y_2_O_3_, 99.995% purity from Stanford Materials Corporation, Lake Forest, CA, USA) using the recrystallization process, including the dissolution in distillated water and ultrapure nitric acid (65%). All nitrates, namely appropriate amounts of Ga(NO_3_)_3_·9H_2_O (Puratronic 99.999% purity from Alfa Aesar, Kandel, GERMANY), Al(NO_3_)_3_·9H_2_O (Puratronic 99.999% purity from Alfa Aesar, Kandel, GERMANY) and Y(NO_3_)_3_ were dissolved in water and mixed together. After that, NH_4_VO_3_ (99% purity from Alfa Aesar, Kandel, GERMANY) were added to the solution. To enable the dissolution of ammonium metavanadate and the complexation of each metal, calculated quantity of citric acid (CA, C_6_H_8_O_7_ with 99.5+% purity from Alfa Aesar, Kandel, GERMANY), used in six-fold excess in respect to the total amount of metal ions, was mixed with all reagents and heated up to 90 °C for 1 h. Next, PEG-200 (poly(ethylene glycol), from Alfa Aesar, Kandel, GERMANY) was added dropwise to the CA-metal complex and stirred for 2 h at 90 °C (CA: PEG-200 was 1:1) to conduct the polyestrification reaction. Then, the resin was obtained by heating at 90 °C for 1 week. In turn, the nanopowders were received via annealing of resin at 1100 °C for 16 h in air atmosphere. 

### 2.2. Characterization

Powder X-ray diffraction (XRD) studies were carried out on PANalytical X’Pert Pro diffractometer equipped with Anton Paar TCU 1000 N Temperature Control Unit using Ni-filtered Cu *Kα* radiation (*V* = 40 kV, *I* = 30 mA). 

Transmission electron microscope images were taken using transmission electron microscopy (TEM) Philips CM-20 SuperTwin with 160 kV of accelerating voltage and 0.25 nm of optical resolution. 

The hydrodynamic size of the nanoparticles was determined by dynamic light scattering (DLS), conducted in Malvern ZetaSizer at room temperature in polystyrene cuvette, using distilled water as a dispersant. 

The emission spectra were measured using the 266 nm excitation line from a laser diode (LD) and a Silver-Nova Super Range TEC Spectrometer form Stellarnet (1 nm spectral resolution) as a detector. The temperature of the sample was controlled using a THMS600 heating stage from Linkam (0.1 1C temperature stability and 0.1 1C set point resolution).

Luminescence decay profiles were recorded using FLS980 Fluorescence Spectrometer from Edinburgh Instruments with μFlash lamp as an excitation source and R928P side window photomultiplier tube from Hamamatsu as a detector.

## 3. Results and Discussion

The yttrium aluminum/gallium garnets crystallize in a cubic structure of Ia3d space group. The general formula of garnets is expressed as follows: A_3_B_2_C_3_O_12_, where three different metallic sites are represented by dodecahedral site (A), octahedral site (B) and tetrahedral site (C), which in our case are occupied by eight-fold coordinated Y^3+^ ions, six-fold coordinated Al^3+^/Ga^3+^ ions and four-fold coordinated Al^3+^/Ga^3+^ ions, respectively. The optically active ions introduced to the structure may occupy different crystallographic sites, which results from the similarities in the coordination number, ionic radii and ionic charge between the host and dopant metal. Therefore, lanthanides (Ln^3+^) prefer to replace A site, while (TM) mainly substitute B and C sites. Additionally, depending on the size of TM ion, they occupy larger (B) (ionic radii 0.67 Å for Al^3+^ and 0.76 Å for Ga^3+^) or smaller (C) (0.53 Å for Al^3+^and 0.61 Å for Ga^3+^) metallic sites. An XRD analysis was used to verify the phase purity of synthesized materials. It is evident that the obtained diffraction peaks of V-doped Y_3_Al_5−x_Ga_x_O_12_ nanocrystals correspond to the reference patterns of cubic structures of adequate host materials (Figure 1a). Observed peaks broadening can be assigned to the small size of the nanoparticles. The *a* cell parameter increases linearly as the Ga^3+^-dopant concentration increased, which results from the enlargement of the crystallographic cell associated with the difference in the ionic radii of Al^3+^and Ga^3+^ ions (rAl^3+^ < rGa^3+^) (Figure 1b). However, it was found that Ga^3+^ ions preferentially occupy four-fold coordinated sites of Al^3+^ rather than the octahedral counterpart. This phenomenon can be explained based on the stronger covalency of Ga^3+^-O^2−^ bonds with respect to the Al^3+^-O^2−^ ones and the lowering of repulsive forces between cations, providing stabilization of the crystal structure [31,32]. On the other hand, the slight shift of the XRD peaks with respect to the reference pattern arises from the implementation of V ions into Y_3_Al_5−x_Ga_x_O_12_ lattice. It was found that Y_3_Al_5−x_Ga_x_O_12_ matrix is a suitable host material for three different V oxidation states, namely V^3+^ and V^5+^ [23,26,33]. The replacement of Ga^3+^ and Al^3+^ ions by V ions is possible due to their comparable ionic radii, which in the case of four-fold coordinated V^5+^ and V^3+^ ions are 0.54 Å, 0.64 Å, respectively, and for six-fold coordinated V^5+^,V^4+^ and V^3+^ ions are 0.68 Å, 0.72 Å and 0.78 Å, respectively. As can be seen from the TEM images, synthesized powders consist of well-crystalized and highly agglomerated nanocrystals (Figure 1c,e,g,i,k). The hydrodynamic sizes of the aggregates of Y_3_Al_5−x_Ga_x_O_12_ nanocrystals examined using DLS analysis were found to be around 300 nm (Figure 1d,f,h,j,l). 

Luminescent properties of V- doped Y_3_Al_5−x_Ga_x_O_12_ nanocrystals were investigated upon 266 nm of excitation in the −150 °C to 300 °C (123.15 K to 573.15 K) temperature range (Figure 2a). The emission spectrum obtained at −150 °C consists of three transition bands, for materials with Ga^3+^ concentration from 1 to 4, and of two emission bands for YGG, being related to the presence of different V oxidations states - V^5+^, V^4+^ and V^3+^. In the course of our previous investigation, it was found that due to the difference in the ionic radii and the charge, V^5+^ ions preferentially occupy surface sites of Al^3+^, while V^3+^ and V^4+^ are mainly located in the core part of the nanoparticles [26,33]. The first broad emission band at 520 nm is attributed to the charge transfer transition of V^5+^(V^4+^ → O^2−^). The second band at 640 nm originates from ^2^E → ^2^T_2_ radiative transition of V^4+^ ions, while the band at 820 nm is associated with ^1^E_2_ → ^3^T_1g_ transition of V^3+^ ions. As can be seen, the addition of Ga^3+^ ions significantly affects the luminescent properties of Y_3_Al_5−x_Ga_x_O_12_:V nanocrystals (Figure 2b). The presented results stay in agreement with the observations obtained for the vanadium doped yttrium aluminum oxide and lanthanum gallium oxide nanoparticles [23,26]. The representative emission spectra measured at −150 °C indicate that the increase of Ga^3+^ concentration caused the enhancement of the V^4+^ emission intensity in respect to the V^5+^ and V^3+^ ones. This effect results from the large ionic radii of V^4+^, which significantly exceeds Al^3+^ ones. Therefore, V^4+^ cannot efficiently replace Al^3+^ in the structure. However, when the concentration of Ga^3+^ ions gradually increases, the number of the crystallographic sites that can be occupied by V^4+^ rises up, leading to the enhancement of ^2^E → ^2^T_2_ emission intensity. Moreover, the Ga-doping induces the reduction of the distance between V^4+^ and V^5+^ ions facilitating the energy transfer between them, which contributed to the V^4+^ luminescent intensity increase. It is worth noticing that the emission of trivalent V dominates in the spectrum up to x = 4, while in the case of YGG V^4+^, the emission band prevails. To quantify these changes the histogram presenting the contribution of the emission intensities (calculated as an integral emission intensity in appropriate spectral range) of particular oxidation state of vanadium ions to the overall emission intensity as a function of Ga^3+^ concentration is presented in Figure 2c. The observed enhancement of V^4+^ emission intensity with respect to the V^5+^ with an increase of Ga^3+^ concentration causes tuning of the emission color toward red emission (Figure 2d). However, for YGG:V, orange emission was found. As has been already proven, the V^5+^ ions are located mainly in the surface part of the nanocrystals [23]. Since the morphology and the size of the nanoparticle is independent on the Ga^3+^ concentration, the number of V^5+^ can be assumed to be constant. The confirmation of this hypothesis is the fact that its lifetime (<τ_V5+_> = 6.4 ms) is independent on the host stoichiometry (Appendix A). On the other hand, the average lifetime of V^3+^ and V^4+^ shortens consequently from 7.6 ms to 7.0 ms and 1.2 ms to 0.5 ms, respectively, with Ga^3+^ concentration (x changed from 1 to 5). 

In order to evaluate how the spectral changes of Y_3_Al_5−x_Ga_x_O_12_ nanocrystal, induced by the stoichiometry modification, affect the performance of analyzed nanoparticles for noncontact temperature sensing, their luminescence spectra were analyzed in a wide range of temperature (from −150 °C to 300 °C) (Figure 3a, Appendix A). In the course of these studies, it was found that emission intensity of each V ion is quenched by temperature; however, their luminescence thermal quenching rates differ (Figure 3b–d). In the case of V^5+^, emission intensity is gradually quenched by almost two orders of magnitude with temperature. However, correlation between Ga^3+^ introduction and temperature of thermal quenching was not observed. This effect is understandable, since, as has been shown before, V^5+^ occupy mainly surface part of the nanoparticles. In turn, the emission intensity of V^4+^ initially decreases with temperature and above some critical temperature, it significantly increases as the temperature grows, which results from the efficient V^5+^ → V^4+^ energy transfer. It was found that the threshold temperature above which rise up of intensity was observed lowers with Ga^3+^ concentration (from around 10 °C for Y_3_Al_4_GaO_12_ to −100 °C for Y_3_AlGa_4_O_12_ and YGG). Additionally the magnitude of the intensity increase growths with Ga^3+^ content. This phenomenon can be explained by the increase of the V^5+^ → V^4+^ energy transfer probability. Higher numbers of Ga^3+^ sites in the structures promote the stabilization of the V^4+^ ions, which, as a consequence, shortens the average distance between V^5+^ and V^4+^ facilitating interionic interactions. Due to the fact that energy of V^5+^ excited state is higher than that of V^4+^, the energy transfer between them occurs with the assistance of the phonon. According to the Miyakava-Dexter theory, the probability of this process is strongly dependent on temperature, which is in agreement with our data [34]. It needs to be noted that although V^5+^ ions serve as a sensitizers for V^4+^, there is no correlation between Ga^3+^ concentration and the V^5+^ luminescence thermal quenching. This comes from the fact that in the case of V^5+^ intensity the luminescence thermal quenching process plays dominant role over V^5+^ → V^4+^ energy transfer. The correlation between Ga^3+^ concentration and the luminescent thermal quenching rate is also evident in the case of V^3+^ ions. The higher the amount of Ga^3+^, the lower the thermal quenching rate of the ^1^E_2_ → ^3^T_1g_ emission band. Above 100 °C, the V^4+^ emission intensity becomes so efficient that its intensity dominates over the V^3+^ ones and thus hinders its emission intensity analysis. In the case of YGG, the V^3+^ emission is impossible to detect. 

Since the emission intensity of V ions in Y_3_Al_5−x_Ga_x_O_12_ nanocrystals is strongly affected by the temperature changes, a quantitative analysis, which verify their performance for non-contact temperature sensing, was performed. For this purpose, the relative sensitivities (*S*) of three different intensity-based luminescent thermometers were calculated according to the following Equation (1):
(1)$$S=\frac{1}{\Omega }\frac{\Delta \Omega }{\Delta T}\cdot 100\%,$$where Ω corresponds to the temperature dependent spectroscopic parameter, which in this case is represented by emission of adequate V ions (S_1_ for V^5+^, S_2_ for V^4+^ and S_3_ for V^3+^), and *ΔΩ* and *ΔT* indicate to the change of Ω and temperature, respectively.

The maximal values of relative sensitivity (S_1_) of V^5+^-based luminescent thermometer, which exceed 2%/°C, were found at temperatures below −100 °C and with increase of temperature S_1_ gradually decreases reaching 1.34%/°C, 1.12%/°C, 1.13%/°C, 1.30%/°C and 0.76%/°C for Y_3_Al_4_GaO_12_, Y_3_Al_3_Ga_2_O_12_, Y_3_Al_2_Ga_3_O_12_, Y_3_AlGa_4_O_12_ and Y_3_Ga_5_O_12_, respectively, in the biological temperature range (0 °C–50 °C). The highest value of the S_1_ was found at −150 °C for Y_3_Ga_5_O_12_, which is in agreement with our expectation that short distance between V^5+^ and V^4+^ facilitates the interionic energy transfer between them. The presented correlations confirm that relative sensitivity of temperature sensors based on V^5+^ emission intensity can be modulated by varying the Ga^3+^-concentration (Figure 3e). In case of Y_3_Al_5−x_Ga_x_O_12_:V^4+^ temperature sensors, the highest value of sensitivity reveal the YGG nanocrystals (S_2max_ = 1.34%/°C at −15 °C), and its value gradually decreases with the lowering of Ga^3+^ concentration. Moreover, the temperature at which maximal S_2_ was found decreases with Ga^3+^ concentration from 75 °C for Y_3_Al_4_GaO_12_ to −15 °C for YGG. This phenomenon is also observed in the case of biological temperature range, where reducing the Ga^3+^ concentration the S value decreases from 1.32%/°C at 0 °C to 0.2%/°C at 30 °C for Y_3_Ga_5_O_12_ to Y_3_Al_4_GaO_12_ (Figure 3f). It should be mentioned here that usable temperature range for this luminescent thermometer (temperature range in which Ω reveals monotonic change) is limited, and the most narrow one was found for YGG (from −100 °C to 120 °C). The negative values of S_2_ come from the fact of the intensity trend reversal. Hence, the balance between relative sensitivity and the usable temperature range can be optimized by the appropriate host material composition. Therefore, depending on the type of application of such luminescent thermometer, including required relative sensitivity and operating temperatures range, different stoichiometry of host material can be proposed. Since the V^3+^ emission intensity monotonically decreases in the temperature range below 200 °C the relative sensitivity S_3_ reveals positive values with the single maxima at temperature which is dependent on the Ga^3+^ concentration (Figure 3g). The increase of Ga^3+^ amount causes the reduction of both value of the S_3_ and the temperature of S max from 1.08%/°C at 152 °C for Y_3_Al_4_GaO_12_ to 0.45%/°C at 51 °C for Y_3_AlGa_4_O_12_.

Although the performance of the intensity-based luminescent thermometer, which take advantage from V^5+^, V^4+^ and V^3+^ emission, are very promising, the reliability of accurate temperature readout is limited due to the fact that emission intensity of a single band may be affected by the number of experimental and physical parameters. Therefore, most of the studies concern the bandshape luminescent thermometer, for which relative emission intensity of two bands is used for temperature sensing. Taking advantage of the fact that emission intensities of V^5+^ and V^4+^ ions reveal opposite temperature dependence, their luminescence intensity ratio (*LIR*) can be used as a sensitive thermometric parameter:
(2)$$LIR=\frac{V^{5+}{(V}^{4+}\to O^{2-})}{V^{4+}{(}^{2}E\to^{2}T_{2})},$$

Analysis of the thermal evolution of *LIR* reveals that for each stoichiometry of the host material the decrease of *LIR*’s value by over three orders of magnitude can be found for −150–300 °C temperature range (Figure 4a). Observed thermal changes of *LIR* significantly exceed those noticed for single ion emission. The relative sensitivities of LIR-based luminescent thermometers (*S_4_*) were defined as follows:
(3)$$S_{4}=\frac{1}{LIR}\cdot \frac{\Delta LIR}{\Delta T}\cdot 100\%,$$

Thereby, the relative sensitivities calculated for LIR-based luminescent thermometers reached values that exceed 2%/°C (Figure 4b). Thermal evolution of *S_4_* attains single maxima at temperature T_Smax_. As was shown before, both the *S_4max_* and T_Smax_ can be successfully modified by the incorporation of the Ga^3+^ ions. The increase of the Ga^3+^ concentration causes the lowering of the T_Smax_ from 20 °C for Y_3_Al_4_GaO_12_ to −100 °C for Y_3_Ga_5_O_12_, while the maximal relative sensitivity increases from 1.47%/°C for Y_3_Al_4_GaO_12_ to 2.48%/°C for Y_3_Ga_5_O_12_ (Figure 4c,d). However, the maximal value of *S_4_* = 2.64%/°C was found for Y_3_Al_2_Ga_3_O_12_. It needs to be mentioned here that, to the best of our knowledge, described nanocrystals reveal the highest values of relative sensitivity for vanadium-based luminescent thermometers up to date. Moreover, it was found that the higher the Ga^3+^ content (Y_3_Al_4_GaO_12_- Y_3_AlGa_4_O_12_), the more significant the change of CIE 1931 chromatic coordinates is (Figure 4e,f).

## 4. Conclusions

In this work, the impact of the host material composition on the temperature-dependent luminescent properties of vanadium-doped nanocrystalline garnets was investigated. It was demonstrated that the incorporation of Ga^3+^ ions into the Y_3_Al_5−x_Ga_x_O_12_:V structure enables modification of the emission color of the phosphor by the stabilization of the vanadium ions on the V^4+^ oxidation state. Taking advantage from the fact that V^4+^ ions, due to their similar ionic radii, mainly occupy the octahedral site of Ga^3+^ ions, the enlargement of their amount leads to the increase of their emission intensity. Moreover, a growing number of V^4+^ ions cause a shortening of the average V^5+^–V^4+^ distance facilitating interionic energy transfer between them. Conducted studies regarding the influence of temperature on the emission intensities of the vanadium ions at different oxidation states reveal that the most susceptible to thermal quenching is the V^5+^ emission intensity. On the other hand, due to the V^5+^ → V^4+^ energy transfer, the V^4+^ emission intensity increases with temperature. The higher the amount of Ga^3+^ ions in the host, the more evident the enhancement of V^4+^ emission intensity and the lower the threshold temperature above which this enhancement occurs. Taking advantage form the fact of opposite temperature dependence of V^5+^ and V^4+^ emission intensities, their ratio was used to create the bandshape luminescent thermometer, to the best of our knowledge, the highest relative sensitivity of V-based luminescent thermometers up to date S_max_, 2.64%/°C, 2.56%/°C and 2.49%/°C for Y_3_Al_2_Ga_3_O_12_ (at 0 °C), Y_3_AlGa_4_O_12_ (at −20 °C) and Y_3_Ga_5_O_12_ (at −100 °C), respectively. With an increase of the Ga^3+^ concentration, the value of the relative sensitivity, as well as the temperature at which *S_max_* was observed, can be modified. Additionally, it was found that the higher the contamination of Ga^3+^ ions, the more evident the change of the chromatic coordinates of emitted light with temperature changes in a −150 °C–300 °C temperature range. As was proven in this manuscript, the introduction of the Ga^3+^ ions in the garnet host enables modification of the performance of nanocrystalline luminescent thermometer like: its usable temperature range, maximal value of the relative sensitivity, as well as the temperature at which maximal sensitivity can be obtained. The dominant effect, which is responsible for described modification of the luminescent properties of V doped luminescent thermometers, is the increase of the V^5+^ → V^4+^ energy transfer probability associated with the growing number of the crystallographic sites that can be occupied by the V^4+^ ions. This shortens the average distance between the interaction ions, facilitating energy transfer process.

## Figures and Tables

**Figure 1 nanomaterials-09-01375-f001:**
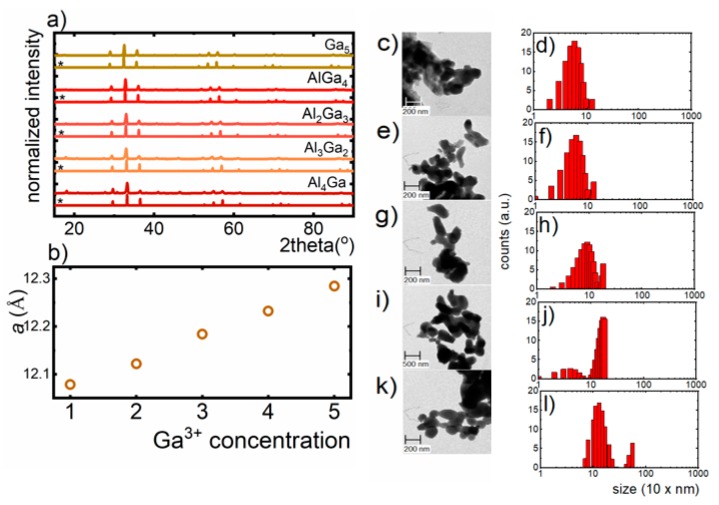
(**a**) XRD patterns of Y_3_Al_5−x_Ga_x_O_12_ nanocrystals, doped with 0.1% V; (**b**) influence of the Ga^3+^ concentration on the *a* cell parameter; (**c**,**e**,**g**,**i**,**k**): the morphology of Y_3_Al_4_GaO_12_, Y_3_Al_3_Ga_2_O_12_, Y_3_Al_2_Ga_3_O_12_, Y_3_AlGa_4_O_12_, Y_3_Ga_5_O_12_, respectively; (**d**,**f**,**h**,**j**,**l**): the distribution of the hydrodynamic size of aggregates.

**Figure 2 nanomaterials-09-01375-f002:**
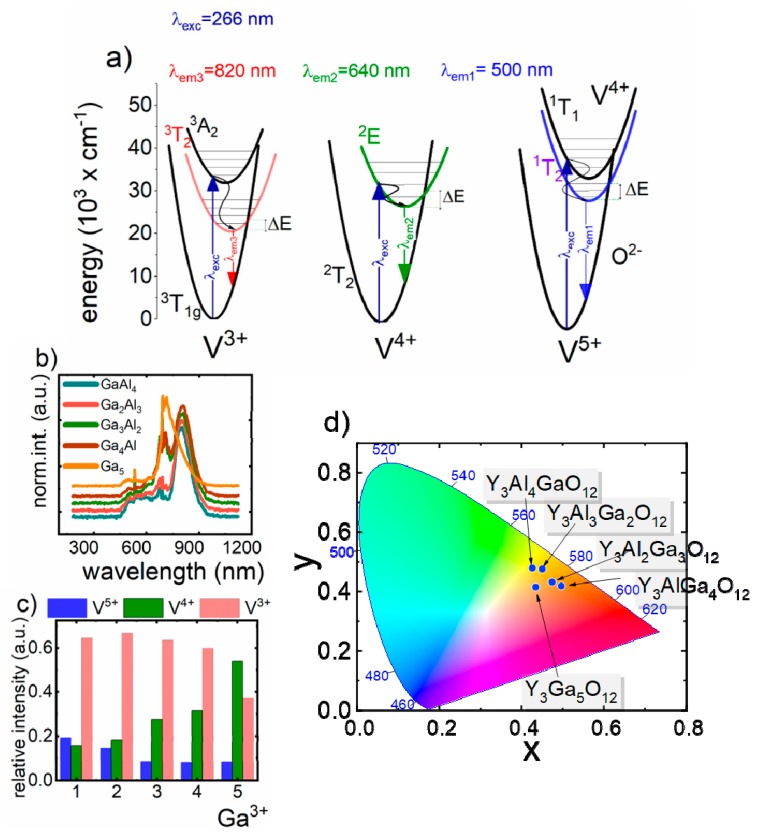
(**a**) The energy diagram of V ions at different oxidation states; (**b**) the influence of Ga-doping on the V emission spectrum (at −150 °C under 266 nm) in Y_3_Al_5−x_Ga_x_O_12_ nanomaterials at 0 °C; (**c**) the contribution of emission intensity of particular oxidation state of V ions into the overall emission spectrum of V-doped Y_3_Al_5−x_Ga_x_O_12_ nanocrystals; (**d**) the Commission internationale de l’éclairage CIE 1931 chromatic coordinates calculated for V:Y_3_Al_5−x_Ga_x_O_12_ nanocrystals at 0 °C.

**Figure 3 nanomaterials-09-01375-f003:**
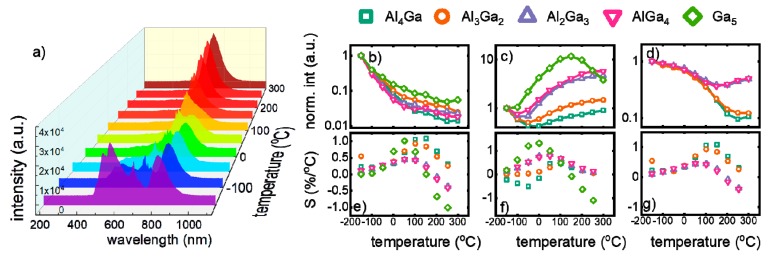
(**a**) Thermal evolution of emission spectrum of Y_3_AlGa_4_O_12_:V nanocrystals; (**b**–**d**) the influence of local temperature on the emission intensity of V^5+^, V^4+^ and V^3+^, respectively; (**e**–**g**) corresponding relative sensitivities.

**Figure 4 nanomaterials-09-01375-f004:**
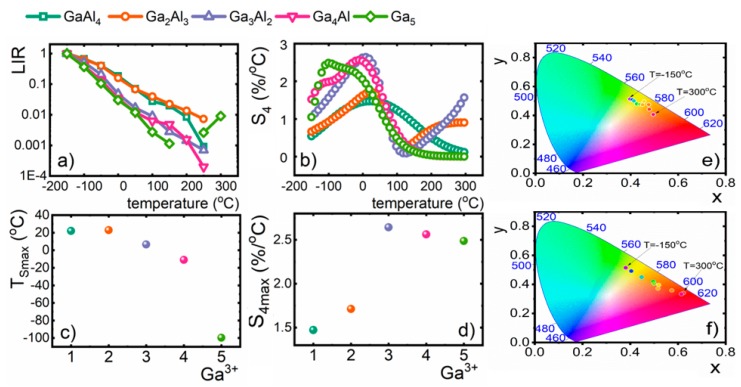
(**a**) Thermal evolution of luminescence intensity ratio (LIR); (**b**) their relative sensitivities for Y_3_Al_5−x_Ga_x_O_12_ nanocrystals; (**c**) the temperature at which the maximal value of S_4_ was observed; (**d**) S_4max_ as a function of Ga^3+^ concentration; (**e**,**f**) the CIE 1931 chromatic coordinates calculated for Y_3_Al_4_GaO_12_:V and Y_3_AlGa_4_O_12_:V nanocrystals, respectively.

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
