# Peer review of "Enhancing the Relative Sensitivity of V5+, V4+ and V3+ Based Luminescent Thermometer by the Optimization of the Stoichiometry of Y3Al5−xGaxO12 Nanocrystals"

_nanomaterials, 2019, doi:10.3390/nano9101375_

Round 1

Reviewer 1 Report

The paper of Marciniak et al., proposes a first investigation on Ga3+ concentration on the emission properties of Y3Al5-xGaxO12: V and their performance for noncontact thermometry. The paper is well written and scientifically sound, while use of a transition metal with multiple valences as a luminescent activator is interesting and highly welcome, considering the abundance of reports on lanthanide based thermometers. Therefore, I recommend the paper for publication following minor revision as detailed below:

Major comment

It is difficult to sustain the nano size of the particles as well as their phase purity starting from Figure 1. Because of their flattened illustration in (a ), the XRD patterns can easily hide the formation of impurity phases. This seems the case of Al4Ga but probably the case is not singular. Further, the authors say (p.3) :” Observed peaks broadening can be assigned to the small size of the nanoparticles: This is hardly observed indeed. At this point, what is the crystallite size estimated by Scherer eq. ? How this is related to TEM images in Figure 1 and hydrodynamic size (X scale seems to be incorrect) ? Please, revise this section. Finally, based on the TEM images in Figure 1, do the authors sustain that the systems are suitable for bioapplications ? Is the synthesis procedure highly reproducible ? From our group experience, obtaining homogenous doping Y3Al5O12 with another transition metal with multiple valences, such as Mo, is not always successful. To this point, I would kindly ask the authors to check again the Synthesis section to see if all relevant details are included. To this point, a brief justification regarding the selection of concentration value of V (0.1%) is needed. 

Minor revision:

Please rephrase : Row 40:“…electrically passive elements what enables achieving the information about i.e. the condition of living organisms where even small temperature fluctuations are usually...”; Rows 105-109; Row 278 “Conducted studies…on the emission intensities of the vanadium ions.. reveals that the most efficiently thermally quenched in the V5+ emission intensity”. Citations needed for Row 37-39: “One of the most important advantage of LT in respect to other temperature measurement techniques is the fact that it provides in real-time temperature readout with unprecedented spatial and thermal resolution.”; Row43 “High spatial resolution of temperature readout is possible by the use of nanoparticle.”; Row 49 - 56: The authors should also mention the host material used, not only the transitional metals. Since (as the authors already stated) an advantage of using TM is “the susceptibility of their optical properties to the modification of the crystal field strength”; Row203: relative sensitivity formula. Please, specify the ionic radii of V3+/4+/5+ Apparently, multiple wrong citations of Figure 4, some of these should be 3 (3g instead of 4g, etc)  please check again.  

Reviewer 2 Report

Authors report on the topic of V based luminescent thermometers based on crystal field engineering in Y3Al5-XGaxO12 nanocrystals synthesised and studied by various methods. The novelty of paper consists of proving that modification of host material composition (Ga ions) allows to tune properties relevant for thermometry. There has been an extensive experimental work to prove hypothesis set by authors. However, in the manuscript there are parts not clearly formulated and citations to references, which do not contain information expected. I feel also that finalization of the manuscript has been in hurry and therefore references to the figures in the text, wording on figures or figure captions are not always correct. Below I indicate my findings and questions to be clarified according to the appearance in the manuscript.

Page 3, line 83: I suggest to explain PEG-200 also like other chemicals have been described.

Page 3, lines 121-123: I found that Ref. 21, is discussing V3+ and V5+ centres, but not V4+. It is a misleading sentence.

Page 4, line 129: Figure 1 a) something is missing; 0.1 % … of what?

Page 4 Figure 1. Is the unit of the third panel correct? d size (x 10-2 nm)

Page 4, line 134: I suggest to show once temperature range in K units as well, typical for low temperature photoluminescence.

Page 4, line 135: The emission spectrum obtained at -150 C consists of THREE TRANSITION BANDS?

Page 4, lines 136-138: Ref. 21 does not describe V4+ centers.

Page 4, line 150-151: Authors write:“Moreover the Ga-doping induces the reduction of the distance between V4+ and V5+ ions facilitating the energy transfer between them, which contributed to the V4+ luminescent intensity increase.” On other hand the amount of V ions in different charge state is strictly connected to the total amount of V ions introduced. If increase of Ga ions facilitates formation of sites favorable for substitution V4+ ions, the concentration of other ions should be diminished. Question is which ones either V5+ or V3+. Next question how relevant is the energy transfer from V5+ states to V4+, if their concentration is rather small ? Authors assume that V5+ concentration is constant (page 5, line 161) based on figure S1. It can also be interpreted that energy transfer is not relevant process, because the decay of 800 nm luminescence is nearly exponential and does not depend on stoichiometry, which leads the reduction of the distance between V4+ and V5+ ions. This issue has to be clarified. Are there any experimental evidences more supporting one of these interpretations?

Page 5, line 159: I found the discussion of surface centres only in Ref. 21.

Page 5, lines 163-164: What is the average lifetime? It is obvious that strong quenching of luminescence occurs. What are the reasons for that?

Page 7, line 180: In my opinion “This effect is understandable since as it was shown V5+ occupy mainly strongly defected surface part of the nanoparticles.” this sentence is not well formulated- strongly defected surface part.

Page 7, lines 88-189: “Due to the fact that energy of V5+ excited state is higher than that of V4+ the energy transfer between then occur with the assistance of the phonon. What about requirements of the overlap of respective states in energy transfer process?

Page 7, line 214: “.. based on V5+ emission intensity, can be modulated via varying the Ga3+-concentration (Figure 4e).”  It seems to be a wrong reference to Figure 4?

Page 8, line 220: “.. from 1.32%/C at 0 C to 0.2%/oC at 30C for Y3Ga5O12 to Y3Al4GaO12 (Figure 4f).”  It seems to be a wrong reference to Figure 4?

Page 9, line 251: What is S4?

Many references are incomplete, even own publications of authors.

This manuscript can be considered for publication only after serious review.

Round 2

Reviewer 2 Report

Honestly saying, I write this review report with very mixed feelings. In one hand authors have responded to the criticism of the reviewers. From that point of view one could accept the paper as it is. But … in several points authors have given a formal answer, not the substantial answer. A typical example of that is :

Page 5, line 159: I found the discussion of surface centres only in Ref. 21.

Author’s response:

The Ref. 23, 26 and 33 also describe the localization of V5+ ions in the surface part of the nanocrystals.

If one does read these papers, one can see that all the experimental justification and discussion to this statement is given in Ref. 23. Refs. 26 and 33 just refer to the paper 23. Authors know in the best way how and there all relevant information is given and there is no point to deliver the absence of any additional information in next papers. It would be even better that if someone else would confirm their findings, but it might take some time.

There is one more serious thing that bothers me. In the review it was pointed that references provided are incomplete. Checking the revised submission, practically the same picture appears. This rises question to the authors have any respect to the Journal requirements, Editor’s and reviewer’s suggestions and work. I shall leave the decision to accept this paper in the present from to EDITOR’s shoulders. Alternative is that the authors team will continue their publication practice and overflow journals with the manuscripts, which do not correspond to expected standards. Upon to Editor’s decision I shall bind also my future participation in the review process of NANOMATERIALS. I believe that there are thousands and thousands of reviewers, including authors of this manuscript, to provide their contributions in order to keep up quality of publications at the highest level.  
